# Targeting the Complement–Sphingolipid System in COVID-19 and Gaucher Diseases: Evidence for a New Treatment Strategy

**DOI:** 10.3390/ijms232214340

**Published:** 2022-11-18

**Authors:** Vyoma Snehal Trivedi, Albert Frank Magnusen, Reena Rani, Luca Marsili, Anne Michele Slavotinek, Daniel Ray Prows, Robert James Hopkin, Mary Ashley McKay, Manoj Kumar Pandey

**Affiliations:** 1Cincinnati Children’s Hospital Medical Center, Division of Human Genetics, 3333 Burnet Avenue, Building R1, MLC 7016, Cincinnati, OH 45229, USA; 2Department of Neurology, James J. and Joan A. Gardner Center for Parkinson’s Disease and Movement Disorders, University of Cincinnati, 3113 Bellevue Ave, Cincinnati, OH 45219, USA; 3Department of Pediatrics, College of Medicine, University of Cincinnati, 3230 Eden Ave, Cincinnati, OH 45267, USA

**Keywords:** lipid, viral infection, rare-genetic disease, innate and adaptive immunity, inflammation

## Abstract

Severe Acute Respiratory Syndrome Coronavirus-2 (SARS-CoV-2)-induced disease (COVID-19) and Gaucher disease (GD) exhibit upregulation of complement 5a (C5a) and its C5aR1 receptor, and excess synthesis of glycosphingolipids that lead to increased infiltration and activation of innate and adaptive immune cells, resulting in massive generation of pro-inflammatory cytokines, chemokines and growth factors. This C5a–C5aR1–glycosphingolipid pathway- induced pro-inflammatory environment causes the tissue damage in COVID-19 and GD. Strikingly, pharmaceutically targeting the C5a–C5aR1 axis or the glycosphingolipid synthesis pathway led to a reduction in glycosphingolipid synthesis and innate and adaptive immune inflammation, and protection from the tissue destruction in both COVID-19 and GD. These results reveal a common involvement of the complement and glycosphingolipid systems driving immune inflammation and tissue damage in COVID-19 and GD, respectively. It is therefore expected that combined targeting of the complement and sphingolipid pathways could ameliorate the tissue destruction, organ failure, and death in patients at high-risk of developing severe cases of COVID-19.

## 1. Introduction

Severe Acute Respiratory Syndrome Coronavirus-2 (SARS-CoV-2)-induced Disease (COVID-19) displays complement activation products (Table 1) and excess formation of sphingolipids [1,2,3,4]. Additionally, SARS-CoV-2 triggers infiltration and activation of several classes of innate and adaptive immune cells, as well as the abnormal production of pro-inflammatory cytokines, chemokines, and growth factors in COVID-19 (Table 1 and Table 2). Such SARS-CoV-2-induced immune inflammation affects multiple organs (i.e., lung, liver, spleen, cardiovascular system, and brain) and causes the development of moderate (e.g., high fever, shortness of breath, loss of taste and/or smell, sore throat, nausea, and vomiting) to severe (e.g., pneumonia, bronchitis, respiratory failure, lung damage and death) symptoms of COVID-19 [5,6,7,8,9,10,11,12,13,14,15,16,17,18,19,20,21,22,23].

**Table 1 ijms-23-14340-t001:** Immune cells and their effector inflammatory mediators in COVID-19.

Coronavirus	Immune Cell Involvement	Source	Changes in Complement Products, Cytokines and Chemokines	References
SARS-CoV-2	Leucocytes, PMNs PCs Endothelial cells	Blood Sera Lungs	C5a ^+++^ C5aR1 ^+++^ MAC ^+++^	[24,25,26,27,28]
SARS-CoV-2	Type-II pneumocytes Pulmonary cells Platelets	Blood Sera Lungs	C1q ^P+++^ C3b-regulatory factor H (FH) ^P+++^ C3 ^P+++^	[26,29,30,31]
SARS-CoV-2	Type-II pneumocytes PBMCs	Blood Sera Lung	C3a ^P+++^ C3aR ^P +++^ C3b-CD46 ^P+++^	[26,31,32,33,34]
SARS-CoV-2	PBMCs	Blood Sera	sC5b-9 ^P+++^	[27,35,36,37]
SARS-2	Respiratory specimen cells Alveolar cells	Blood Sera Lung	C4d ^P+++^	[35,38]
SARS-CoV-2	Respiratory specimen cells	Blood Sera	C3bBbP ^P+++^	[35]
SARS-CoV-2	Respiratory specimen cells	Blood Sera	C3bc ^P+++^	[35]
SARS-CoV-2	PBMCs Glomeruli Cardiac Microthrombi and Alveolar cells	Blood Sera Lung Heart Kidney	C5b-C9 ^P+++^	[36,37,38,39,40]
SARS-CoV-2	Pulmonary cells	Lung	C1r ^P+++^	[29]
SARS-CoV-1	Pulmonary cells	Lung	iC3b ^P+++^	[31]
SARS-CoV-1	Pulmonary cells	Lung	C3c ^P++^	[31]
SARS-CoV-1	Pulmonary cells	Lung	C3dg ^P+++^	[31]
MERS	MOs/Mɸs T cells mDCs pDCs PMNs	Liver SPL Lung Kidney Heart Serum	IL12 ^M+++^ IL8 ^M+++^ TNFα ^M+++^ IL-6 ^M&P+++^ IFNλ ^M+++^ CXCL10 ^M&P+++^ CCL2 ^M+++^ CCL3 ^M+++^ CCL5 ^M+++^ IFNα ^M+ & P+++^ IFNβ ^M+ & P+++^	[41,42,43,44,45,46,47]
SARS-CoV-1	Mos Mɸs DCs Cord blood cells	Lung Blood	IFNβ ^M+^ IFNα ^M+^ TNFα ^M+++^ IFNλ ^M+++^ IL8 ^M+++^ TNFα ^M+++^ IL6 ^M+++^ CCL2 ^M+++^ CCL3 ^M+++^ CCL5 ^M+++^ CXCL10 ^M+^	[42,48,49,50]
SARS-CoV-2	MOs Mɸs	Blood	IL6 ^P+++^ IL8 ^P+++^ IL10 ^P+++^ TNFα ^P+++^	[51]
SARS-CoV-2	CD8^+^ cells NK^+^ cells	PBMCs	IL2 ^P+^ TNFα ^P+^ IFNγ ^P+++^ Granzyme B ^P+++^	[52,53]
SARS-CoV-2	CD8^+^ cells CD4^+^ cells	Liver Lung Heart	Cytokines ^NR^	[53]
SARS-CoV-2	PMNs	Lung	Cytokines ^NR^	[54]
MERS	CD4^+^ T cells CD8^+^ T cells	Lymph Nodes Spleen Tonsils PBMCs	Caspase-3 ^P+++^	[55]
SARS-CoV-1	CD4^+^ cells CD8^+^ cells CD45RO^+^ and CD27^+^ cells	PBMCs	IL2 ^P++^ TNFα ^P++^ IFNγ ^P+++^ IL4 ^PNS^ CXCL10 ^PNS^	[56,57]
SARS-CoV-2	CD4^+^ T cells CD8^+^ T cells Regulatory T cells	PBMCs	CCR6 ^P+++^ Perforin ^P+++^ IL6 ^P+++^ IL2 ^M+++^ IL7 ^M+++^	[51,58,59,60,61,62,63]
MERS	Epithelial cells	Lung	IL1β ^M+++^ IL6 ^M+++^ IL8 ^M+++^ IFNα ^M+^ CCL2 ^M+^ CXCL10 ^M+^	[64,65]
SARS-CoV-1	Epithelial cells	Lung	TNFα ^M+++^ IFNβ ^M+++^ CXCL10 ^M+++^	[64]

COVID-19 (Coronavirus disease 2019), MERS-CoV (Middle East Respiratory Syndrome Coronavirus), SARS-CoV-1 (Severe Acute Respiratory Syndrome Coronavirus-1), SARS-CoV-2 (Severe Acute Respiratory Syndrome Coronavirus-2), AECs (Airway epithelial cells), Mɸs (Macrophages), DCs (Dendritic cells), PMNs (Polymorphonuclear cells), NK (Natural Killer cells), IFN (interferon), α (alpha), β (beta), γ(gamma), λ (lambda), IL (interleukin), TNF (tumor necrosis factor), CCL (Chemokine (C-C motif) ligand), CXCL (Chemokine (C-X-C motif) ligand), PBMCs (Peripheral Blood Mononuclear Cells), PCs (Pulmonary Cells), MAC (Membrane Attack Complex; MAC), C3a (Complement 3a), C3b (Complement 3b), C3aR (Complement 3a Receptor), C5a (Complement 5a), C5aR1 (C5a Receptor 1), sC5b-9 (soluble C5b-9; MAC); C4d (Complement 4d). C3bBbP (complement 3 b bound protease fragment), C3bc (Complement 3 bc), P (Protein expression level), M (mRNA expression level) + (low), ++ (moderate), and +++ (high), NS (not significant).

Increased levels of complement activation products have been linked to innate and adaptive immune cell activation and increased production of pro-inflammatory cytokines, chemokines, and growth factors in GD (Table 3 and Table 4). The excess tissue and cellular accumulation of glucosylceramides (GCs) and their subsequent roles in the induction of innate and adaptive immune inflammation significantly affect visceral organs (e.g., liver, spleen, lung, bone, and kidney) and the central nervous system (CNS), causing the development of GD manifestations characterized by anemia, thrombocytopenia, hypergammaglobulinemia, splenomegaly, hepatomegaly, respiratory distress, skeletal weakness, loss of neurons, and death [66,67,68,69,70,71,72,73,74,75,76].

Clinical evidence together with laboratory investigations suggest a common involvement of the C5–C5a–C5aR1 and glycosphingolipids (GSLs) pathways for activating innate and adaptive immune cells, including monocytes (MOs), macrophages (Mɸs) dendritic cells (DCs), polymorphonuclear cells (PMNs), and CD4^+^ T cells. Similarly, the abnormal production of pro-inflammatory cytokines was identified, including interferon-alpha (IFNα), IFN-gamma (IFNγ), tumor necrosis factor-alpha (TNFα), interleukin-1 (IL1), IL2, IL6, IL7, IL8, IL12, IL17), CCL (C-C motif ligand chemokines, e.g., CCL2, CCL3, and CCL5), CXCL (C-X-C motif ligand chemokines, e.g., CXCL9 and CXCL10), and growth factors (e.g., transforming growth factor-beta, TGFβ), granulocyte colony stimulating factor (GCSF), granulocyte-Mɸ colony stimulating factor (GMCSF) in COVID-19 and GD (Table 1, Table 2, Table 3 and Table 4). Such immune abnormalities affect multiple tissues and lead to the development of organ failure and/or death in both COVID-19 and GD patients [6,7,8,9,10,11,12,13,14,71,72,74,75,76,77].

Several vaccines and alternative drugs have been applied and/or proposed for treating COVID-19 and its complications (Table 5). Some of these treatments have been used off-label, though their efficacy is still debated. Studies have shown that pharmaceutical targeting of complement at the level of the C5–C5a–C5aR1 axis or at glucosylceramide synthase (GCS) inhibited the replication of SARS-CoV-2, and the SARS-CoV-2 or GC-induced immune inflammation and tissue destruction in COVID-19 and GD [78,79,80,81,82]. The study therefore provided updated information regarding the involvement of the complement-sphingolipid axis and the impact in propagating the immune inflammation and disease process of COVID-19 and GD. This information could be helpful to understand the disease mechanism of COVID-19 and to aid in the development of additional potential therapies to slow or reduce disease severity and death in COVID-19 patients. 

### 1.1. COVID-19

COVID-19 is caused by infection with SARS-CoV-2, a member of the family of betacoronaviruses that also includes the SARS-CoV-1 and Middle East Respiratory Syndrome-CoV (MERS-CoV) [83,84]. SARS-CoV-2 is a large, enveloped, single-stranded positive-sense RNA virus with a genome size of about 30 kb. The 5′ end of the SARS-CoV-2 genome encodes two polyproteins termed PP1a and PP1ab, which are mutually called replicases. These polyproteins are classified with 16 non-structural proteins, including RNA-dependent RNA polymerase (RDRP), 3-chymotrypsin-like protease (3CLP), and papain-like protease (PLP). The 3′ end of the SARS-CoV-2 genome encodes four essential structural proteins: spike (S; essential for the viral entry into host cells), envelope (E; responsible for viral membrane twist and binding to the nucleocapsid), membrane (M; bind to the viral RNA genome and guarantee the conservation of the RNA in the shape of the beads-on-a-string) and the nucleocapsid (N; required for viral replication and pathogenesis of the disease). The 3′ end also encodes non-structural proteins, including PLP, 3CLP, RDRP, helicase, and the collection of accessory proteins, which affect the host-specific immune reactions [85,86,87]. The S protein is critical for the infectivity of SARS-CoV-2 and is cut by a host protease into the S1 and S2 subunits. The S1 subunit binds to angiotensin converting enzyme-2 (ACE2), which acts as a protease to cleave angiotensin II and also counteracts the effect of angiotensin II [88,89]. ACE2 is the cellular receptor for SARS-CoV-2 and is widely expressed in blood vessels, tongue, lung, adipose tissue, adrenal gland, heart, esophagus, lung, muscle, ovary [22,90,91,92] and eye tissues (i.e., conjunctiva, choroid, vascular endothelium, and nerves) [93,94,95,96,97]. 

The S2 subunit is activated by transmembrane protease serine-2 (TMPRSS2) associated with the host surface. These combined actions result in host-viral membrane fusion and SARS-CoV-2 entry into the host cells [98,99]. The viral RNA genome is released into the host cell cytoplasm, where it first uses the host translational machinery for the formation of the viral structural and accessory proteins [85,99]. The newly synthesized structural and accessory proteins are transferred through the endoplasmic reticulum and Golgi bodies followed by assembly of new virions in the growing Golgi vesicles [87]. Finally, similar to the MERS-CoV and SARS-CoV-1, the mature SARS-CoV-2 virions are exocytosed from the host cell into the surrounding environment to repeat the infection cycle. Infection results in activation of innate and adaptive immune cells and a massive generation of several pro-inflammatory mediators (Table 1 and Table 2) that instigate moderate to severe disease symptoms of COVID-19 [14,15,16,17,18,19,20,21,22,23]. The SARS-CoV-2-induced development of COVID-19 disease has been reported in people with immunocompromised and morbid conditions, such as sepsis, acute cardiac injury, heart failure, and multi-organ (e.g., liver, spleen, kidney, and brain) disease [100]. Epidemiological studies related to COVID-19 have found that males are slightly more prone to infection as compared to females, but both sexes experience severe forms of COVID-19 and death. Persons > 60 years old or with chronic diseases, such as type 2 diabetes and essential hypertension, were at higher risk for SARS-CoV-2-induced systemic inflammation leading to a severe form of COVID-19 [101,102,103].


**Table 2 ijms-23-14340-t002:** Coronavirus-induced production of circulatory cytokines.

Coronavirus	Source	Cytokines	Chemokines	Growth Factors	References
MERS	Sera	IFNα ^P+++^ IL6 ^P+++^ IL8 ^P+++^	CCL5 ^P+++^ CXCL10 ^P+++^		[45,47,104]
SARS-CoV-1	Sera	IFNα ^P+++^ IFNγ ^P+++^ IL1 ^P+++^ IL2 ^P+++^ IL6 ^P+++^ IL8 ^P+++^ IL10 ^P+^ IL12 ^P+++^	CCL2 ^P+++^ CXCL9 ^P+++^ CXCL10 ^P+++^	TGFβ ^P+++^	[104,105,106,107,108,109,110,111,112,113,114,115]
SARS-CoV-2	Sera	IFNγ ^P+++^ TNFα ^P+++^ IL1b ^P+++^ IL1RA ^P+++^ IL2 ^P++^ IL2R ^P++^ IL4 ^P++^ IL5 ^P++^ IL6 ^P+++^ IL7 ^P+++^ IL8 ^P+++^ IL9 ^P+++^ IL10 ^NS^ IL12 ^P+++^	CCL2 ^P+^ CCL3 ^P+^ CXCL10 ^P+^	G-CSF ^P+++^ GMCSF ^P+++^ VEGF ^P+++^ FGF ^P+++^ PDGF ^P+++^	[57,59,62,104,116,117,118,119,120]
SARS-CoV-2	Sera		CCL2 ^P+++^		[115,120]
SARS-CoV-2	Sera		CCL3 ^NS^		[115]
SARS-CoV-2	Sera		CXCL8 ^P+++^		[120]
SARS-CoV-2	Sera		CXCL10 ^P++^		[115,120]

MERS-CoV (Middle East Respiratory Syndrome Coronavirus), SARS-CoV-1 (Severe Acute Respiratory Syndrome Coronavirus-1), SARS-CoV-2 (Severe Acute Respiratory Syndrome Coronavirus-2), IFN (interferon), IL (interleukin), TNF (tumor necrosis factor), CCL (C-C motif ligand), CXCL (C-X-C motif ligand), TGF (transforming growth factor), GCSF (granulocyte colony stimulating factor), GMCSF (granulocyte-Mɸ colony stimulating factor), VEGF (vascular endothelial cell growth factor), FGF (fibroblast growth factor), PDGF (platelet-derived growth factor), α (alpha), β (beta), γ (gamma), λ (lambda), P (Protein expression level), M (mRNA expression level): + (low), ++ (moderate), and +++ (high), NS (not significant).

**Table 3 ijms-23-14340-t003:** Immune cells involvement in GD.

	Mouse Model of GD		GD Patients	
Immune Cells	Tissue Recruitment	References	Immune Cells	References
MOs	Blood +++	[121]	Blood ^-^	[122,123]
Mɸs	Blood +++, Liver +++, Spleen +++, Lung +++	[66,121,124,125]	Lymph node +++	[126]
mDCs	Blood +++, Liver +++, Spleen +++, Lung +++	[66,121,124,125]	Blood ^-^	[122,123,127,128]
pDCs			Blood ^-^	[122,123,127]
PMNs	Blood +++, Liver +++, Spleen +++, Bone Marrow +++	[66,121]		
CD4 + TCells	Liver +++, Spleen +++, Lung +++	[66,124,125]	Blood +++	[127,128]
CD8 + T Cells	Thymus +++, Spleen +++	[124,125]	Blood +++	[128,129]
NK Cells			Blood ^-^	[127,129]

MOs (Monocytes), Mɸs (Macrophages), mDCs (myeloid dendritic cells), pDCs (plasmacytoid dendritic cells), PMNs (Polymorphonuclear cells), NK (Natural killer cells), Increased (+++) and decreased (-) tissue recruitment.

### 1.2. COVID-19 Vaccines and Alternative Treatments

Several vaccines have been developed against multiple strains of SARS-CoV-2 infection, including mRNA vaccines, viral vector vaccines, inactivated vaccines, and protein-based vaccines [130] (Table 5). mRNA vaccines have shown an efficacy ranging from 48 to >90% at 5–6 months’ follow-up post second vaccine dose [130]. Viral vector vaccines had an efficacy ranging from 65 to 91.6% (Sputnik), around 70% (in Brazil), or between 10 and 64% (South Africa), according to the different vaccine types and countries in which they were used [130]. Finally, inactivated and subunit vaccines, showed an efficacy ranging from 50 to 91.6%, again based on the different subtypes and the country in which they were employed [130]. SARS-CoV-2 continuously develops mutations (variants) and will continue to do so because relatively few people globally have been vaccinated [131]. These findings suggest that the variability in vaccine efficacy is linked to the dominant variant circulating in the country at the time.

Despite the high efficacy of different vaccines, multiple studies have shown the phenomenon of immunity waning occurring approximately 6 months after vaccine administration [132]. In addition, all of the vaccines can be associated with severe adverse events, namely severe anaphylaxis, myocarditis/pericarditis, lymphadenopathy, appendicitis, herpes-zoster infection, viral facial palsy, transverse myelitis, Guillain-Barré syndrome, encephalopathy, thromboembolism, and thrombosis-thrombocytopenia syndrome [130]. These severe adverse events are rare, and may vary significantly according to specific features of the target population such as age. For example, myocarditis is more frequently reported in males under the age of 30 when compared to other population categories [133].

Several drugs, mainly belonging to the classes of antivirals, antibiotics, immune-modulators, immune-suppressants, or anti-inflammatory molecules, have also been proposed for treating COVID-19 and its complications (Table 5). Some of these drugs have been used off-label and their efficacy is still debated. Lopinavir and ritonavir are two protease inhibitors used for the treatment of human immunodeficiency virus (HIV)-induced acquired immunodeficiency syndrome (AIDS) [134,135]. Their use in combination has been proposed for COVID-19 [135,136]. These drugs may also prevent the secondary immune burst without causing severe adverse events at early stages of COVID-19 and are therefore not recommended for COVID-19 at severe stages of the disease [137]. Among several repurposed drugs currently used in patients with COVID-19, the nucleoside analogs are a preferred class [138], with two drugs in particular: (1) the nucleotide precursor favipiravir, and (2) the nucleoside analogue remdesivir, which inhibit the RDRP [139,140]. A recent systematic review of the literature has shown that favipiravir may significantly improve the clinical condition of COVID-19 patients, although the results suggested no significant differences in the length of hospitalization and clinical recovery [138]. A combination of favipiravir with other therapies (e.g., tocilizumab) has shown more promising results in improving a patients’ clinical status [138]. Possible safety concerns for favipiravir include increased blood uric acid and teratogenicity, as the medication is secreted in semen and breast milk and has demonstrated toxicity in animal studies [141]. Remdesivir may help improve recovery and the clinical outcomes of hospitalized COVID-19 patients, although its effects in reducing mortality are still uncertain [142,143]. It should be administered in a hospital setting. Unfortunately, data on its possible toxicity are sparse [138,144]. 

Several immune-modulating agents have been proposed for treating COVID-19 with the aim of reducing SARS-CoV-2–induced cytokine signalling and increased inflammation (Table 5). Among the specific immune-modulator drugs, the most important are IFNγ, GMCSF, IL1 receptor antagonists, IL-6 receptor antagonists, Janus kinase (JAK) inhibitors, and monoclonal antibodies to TNFα and GMCSF [145,146,147,148,149,150]. Among the non-specific immune modulator drugs, the most important are intravenous immunoglobulins (IVIGs), corticosteroids, and IFNβ-1b or α-2b that have antiviral and immunomodulating properties [149,151,152]. Finally, among immune-modulators with mixed effects, the most important are statins, renin-angiotensin-aldosterone system (RAAS) inhibitors, angiotensin-converting-enzyme (ACE) inhibitors, angiotensin-receptor blockers, and macrolides [153,154]. All these drugs may be associated with side effects of variable immune system suppression and may result in opportunistic infections.

### 1.3. SARS-CoV-2 Reinfection

Several cases of SARS-CoV-2 reinfection after vaccination/first infection have been reported in association with waning of the immune response. While reinfection with viruses associated with systemic infections is uncommon, reinfection with viruses causing mucosal infections without viremia, such as respiratory syncytial virus, influenza, and coronavirus, is much more common [155]. In unvaccinated patients infected by coronaviruses, the mean time to reinfection ranged from 30 to 55 months, and 14% of patients infected had multiple reinfections with the same seasonal coronavirus strain [156]. In a more recent systematic review of the literature on more than 615,000 participants, reinfection was found to be an uncommon event and occurred in 0–1.1% [157]. The data from that review suggested that overall, naturally acquired SARS-CoV-2 immunity does not wane for about 10 months post-infection [157].

### 1.4. Gaucher Disease

GD is a classic example of a lysosomal storage disease and has a worldwide incidence of approximately 1/40,000 to 1/60,000 [158]. GD is classified into Types 1, 2, and 3, which are all caused by pathogenic variants in *GBA1* (in human)/*Gba1* (in mouse) that lead to the functional disruption of the encoded lysosomal enzyme, acid β-glucosidase (β-D-glucosyl-N-acylsphingosine glucohydrolase, EC 4.2.1.25; GCase) and to excess tissue accumulation of glucosylceramides (GC) [23]. Type 1 GD mainly affects the liver, spleen, lung, bone, and kidney [77,159]. A significant proportion of patients with Type 1 GD also have CNS manifestations characterized by the mild brain inflammation [66,67,160,161,162]. Type 2 and Type 3 GD are the neuronopathic forms of GD (nGD), which are characterized by severe and chronic brain inflammation that leads to the loss of neurons and early death (e.g., <2 years in patients with Type 2 nGD and 10–40 years in patients with Type 3 nGD) [162,163,164,165,166,167,168,169,170,171,172,173,174,175]. The main neurological symptoms of Type 2 and 3 nGDs are characterized by selective degeneration of the cerebellar dentate nucleus and the dentato-rubro-thalamic pathway, generalized epilepsy and seizures, horizontal saccadic eye movements, ataxia, spasticity, oculomotor abnormalities, hypertonia of the neck muscles, extreme arching of the neck, bulbar signs, limb rigidity, occasional choreoathetoid movements, and progressive dementia [176,177,178,179,180,181,182,183,184,185,186,187,188,189]. Enzyme replacement therapy (e.g., imiglucerase, velaglucerase, or taliglucerase) and substrate reduction therapy (e.g., eliglustat and miglustat) are available to treat many of the visceral aspects of GD but have no impact on the CNS disease and are of limited benefit in the management of immune inflammation and disease complications in multiple organs such as the bones, lungs, and lymph nodes [70,190,191,192]. The development of alternative therapies, i.e., gene, substrate reduction, and enzyme replacement therapies, has been hampered by limitations in understanding disease pathogenesis, inability of therapies to fully cross the blood-brain barrier, and toxicity concerns due to procedural risks [159,193,194,195,196].

### 1.5. Complement Activation in COVID-19 and Gaucher Disease

The complement system comprises a group of liquid and cell membrane-associated proteins, which are mainly produced in the liver. However, certain brain cells, such as microglial cells, astrocytes, and neurons, are also involved in the direct synthesis of complement proteins [197,198,199]. Complement activation is a complex process, which largely occurs by the classical, alternative, and lectin pathways [200]. The classical pathway is activated by ligation of the IgG and/or IgM immune complexes (ICs) to their corresponding receptors and/or C1q on the cell surface. The alternative pathway is activated by binding of spontaneously activated C3 protein (C3b fragment) to host and non-host cell surfaces [201,202]. The lectin pathway is activated by the binding of the mannan-binding lectin (MBL) to mannose-containing carbohydrates or related ficolins to certain carbohydrates or acetylated structures [198,203,204]. Each of the complement activation pathways follows a series of reactions generating common key components known as C3 and C5 [205]. The downstream cleavage of C3 by the C3 convertases causes the formation of C3a and C3b [205]. Similarly, the downstream cleavage of C5 by the C5 convertases causes the formation of C5a and C5b [205]. C3a binds the C3aR receptor, and C5a binds C5aR1 and C5aR2 receptors [205]. C3b is a major opsonin that induces the tagging and phagocytic uptake of pathogens, and C5b initiates the terminal complement pathway, resulting in the formation of the membrane attack complex (MAC) composed of C5b, C6, C7, C8 and multiple C9 molecules [35,205,206,207,208]. Several activated components of the complement system are essential for controlling cellular and metabolic functions in both visceral organs and the CNS [209,210,211,212,213]. However, the abnormal activation and production of complement components such as C3, C3b, iC3b, CR3, C5, C5a, and MAC/C5b-9 can contribute to visceral and CNS tissue damage in several diseases including, but not limited to, allergic diseases, cardiovascular diseases, age-related macular degeneration, systemic lupus erythematosus, traumatic brain injury, stroke, neuromyelitis optica spectrum disorders, and neurodegenerative diseases such as amyotrophic lateral sclerosis, Alzheimer’s disease and Huntington disease [214,215,216,217,218,219,220,221,222,223,224,225,226,227,228,229,230,231,232].

Complement activation has been observed in H5N1 and H1N1 influenza virus infections [233]. The sera and lung tissues of the MERS-CoV-infected, hDPP4-transgenic (hDPP4-Tg) mouse model showed complement activation and increased production of C5a and C5b-9, as well as increased viral replication [234,235]. Furthermore, targeting C5aR with anti-C5aR antibody treatment in the MERS-CoV-infected hDPP4-Tg mouse model decreased viral replication and damage to lung and spleen tissue [234,235]. The lung of the mouse-adapted SARS-CoV animal model has also demonstrated complement activation and higher virus replication [31]. To investigate whether SARS-CoV triggered complement activation in the mouse model, C3^-/-^ and background matched control wildtype (WT) mice were infected with SARS-CoV-2 and their lung tissues analyzed for immune inflammation. The data showed that C3^-/-^ mice infected with SARS-CoV-2 were protected against the development of respiratory dysfunction and lung damage when compared to WT mice [31].

SARS-CoV-2 infected primary human airway epithelial cells also caused complement activation and increased production of C3a [236]. C3aR and C5aR targeting in cellular models also showed protection from lung tissue damage [236]. Lung and skin tissues and sera from patients with severe COVID-19 have shown increased complement activation and massive generation of C5a, C5b-C9, and C5aR1 in blood and pulmonary myeloid cells [209]. Studies have shown that SARS-CoV-2 induced hyperinflammation is linked to the development of acute respiratory distress syndrome (ARDS), systemic clotting, and a variety of cutaneous manifestations in patients with COVID-19 [80,81,82,237,238,239,240,241]. Strikingly, administration of C3 (AMY101) or C5 (eculizumab) targeting drugs have shown marked reduction in the SARS-CoV-2-induced development of severe disease symptoms and lung damage in COVID-19 patients [80,81,82]. These data suggest that complement activation and the resultant activation of the C5a–C5aR1 axis is critical for propagating the disease complications in patients with COVID-19.

Investigations into the mechanisms by which complement activation occurs in COVID-19 revealed (1) the receptor-binding domain of the spike protein of SARS-CoV-2 binds to specific IgG/IgM antibodies, (2) heparan sulfate binds to the spike protein of SARS-CoV-2, and (3) recognition of the spike and nucleocapsid proteins of SARS-CoV-2 by lectin pathway components subsequently triggers complement activation by the classical, alternative, and lectin pathways [32,35,242,243,244]. Mouse IgGs (IgG1, IgG2a/c, IgG2b, and IgG3) and their matching receptors (e.g., FcγRI, FcγRIIb, FcγRIII and FcγRIV) differ from the human IgGs (e.g., IgG1, IgG2, IgG3, and IgG 4) and their receptors (e.g., FcγRI, FcγRIIa, FcγRIIc, FcγRIIIa, FcγRIIb, and FcγRIIIb) [245]. Human or mouse FcγRI binds only to the monomeric IgG, but all the other FcγRs in the mouse (e.g., FcγRIIb, FcγRIII and FcγRIV) and human (e.g., FcγRIIa, FcγRIIb, FcγRIIIa and FcγRIIIb) can bind to IgG-ICs [245,246]. SARS-CoV-2 infection can result in the sustained development of memory B cells and long-lived bone marrow plasma B cells [247,248,249] and the production of IgG, IgM, and IgA antibodies to S, the RBD of S, and N proteins (Table 6). The SARS-CoV-2-specific IgA/IgM antibodies are present only for a short period of time, whereas the SARS-CoV-2-specific IgG antibodies persist for long periods of time, thereby providing protection against SARS-CoV-2 in patients with COVID-19 [250,251].

The crosslinking of IgG2a/IgG2b ICs to the FcγRIII/FcγRIV in mice and IgG1 IC crosslinking to the activating FcγR in humans cause optimal complement activation and C5a formation [23,252,253,254,255,256,257,258]. Analysis of IgG and their receptors revealed higher expression of the activating FcγR receptors [259] and increased production of SARS-CoV-2-specific IgG antibodies (IgG1 > IgG2 > IgG3) in patients with COVID-19 [23,251,260]. These findings suggest that the IgG isotypes developing in COVID-19 contribute to complement activation. We and others have reported higher levels of complement activation products (e.g., C1q, C3, C4b C5a, C3aR, C5aR1, and C-type lectins) in mouse models and patients with GD (Table 4). Studies on the *Gba1* mutant mouse model, specimens from Type 1 GD patients, and induced pluripotent stem cell (iPSC)-derived Mɸs and lymph nodes from Type 2 and type 3 nGD patients have shown that activation of complement and the C5a–C5aR1 axis leads to tissue inflammation and organ failure in GD [126]. Further, we and others have shown that genetic deficiency of C5aR1 or pharmaceutical targeting to block this receptor in mouse models of GD and in human cell models resulted in marked reduction of immune inflammation and tissue damage (e.g., lung, liver, and spleen) [79,258]. Increased production of autoantibodies specific to GC, IgG2a/c and IgG2b are seen in the *Gba1^9V/-^* mutant mouse model of GD. Also, high levels of GC-specific IgG1 autoantibodies and lower levels of IgG2 and IgG3 autoantibodies have been observed in GD patients [258]. Similarly, experimental mouse models of GD and GD patients have shown elevated levels of mouse/human specific FcγRIII and FcγRIV [258,261,262]. C1q-binding to IgG-IC drives a series of events resulting in the proteolytic cleavage of C4, C2, and C3 and eventually of C5 into C5a and C5b. Further, C5a can be generated locally in immune cells through an FcγR-dependent mechanism involving LAT phosphorylation and production of a C5-cleaving protease [215,252,253,254,255,256,257,263,264,265]. We have observed that massive GC accumulation drives the production of GC-specific IgG2a/c and IgG2b-ICs in *Gba1^9V/-^* mice, as well as IgG1 and IgG3 ICs in GD patients to initiate complement activation through the classical pathway [258]. These findings suggest common involvement of complement activation and C5a–C5aR1 axis-mediated tissue inflammation in GD and COVID-19. Therefore, targeting complement at the level of the C5a–C5aR1 axis could control SARS-CoV-2-induced tissue inflammation in COVID-19 in addition to GD patients with COVID-19.

**Table 4 ijms-23-14340-t004:** Immune cells and their effector inflammatory mediators in GD.

Immunological Signature	Mouse Model of GD		Human GD	
	Source	References	Source	References
C1qα ^M+++^	Lung	[261]		
C1qβ ^M+++^	Lung	[261]	Lymph node	[126]
C1qc ^M+++^	Lung	[261]		
C3 ^M+++^			Lymph node	[126]
Clec4d ^M+++^ Clec4n ^M+++^ Clec5a ^M+++^	Lung	[261]		
Clec7 ^M+++^	Liver	[261]		
MR ^P+^			Lymph node, Mɸs	[126]
C3aR1 ^M+++^	Lung	[261]		
C4b ^P++^			Lymph node	[126]
C5a ^M/P+++^	Mɸs, DCs, Sera	[258]	Lymph node, CBE-induced GCase-targeted Mɸs, iPSCs-derived Mɸs of human Gaucher disease, Sera	[79,126,258]
C5aR1 ^P+++^	Mɸs and DCs	[258]	Lymph node, Mɸs	[126,258]
IFNβ ^P+++^	Neurons	[266]		
IFNγ ^P+++^	Anti-CD3 and CD28-stimulated liver, spleen, and lung derived CD4^+^ T cells, GC-stimulated liver, spleen, and lung-derived DCs, CD4^+^ T cells, sera, plasma, microglial cells	[66,267]	Sera/Plasma and GC-ICs stimulated CBE-induced GCase targeted Mɸs	[66,67,267]
TNFα ^P+++^	Anti-CD3 and CD28-stimulated Liver, spleen, and lung-derived CD4^+^ T cells, GC-stimulated liver, spleen, and lung-derived DCs, CD4^+^ T cells, sera, plasma, microglial cells, whole brain	[66,267]	Sera/Plasma, GC-ICs stimulated CBE-induced GCase-targeted Mɸs, and iPSCs-derived Mɸs of human Gaucher disease	[66,67,79,267]
IL1α ^M/P +++^	Sera/Plasma and whole brain	[258]		
IL1β ^P+++^	Anti-CD3 and CD28-stimulated liver, spleen, and lung-derived CD4^+^ T cells, GC-stimulated liver, spleen, and lung-derived DCs, CD4^+^ T cells, sera, plasma, and whole brain	[66,258,267]	Sera/Plasma, GC-ICs stimulated CBE-induced GCase-targeted Mɸs	[66,67,267]
IL6 ^M/P+++^	Anti-CD3 and CD28-stimulated liver, spleen, and lung- derived CD4^+^ T cells, GC-stimulated liver, spleen, and lung derived DCs, CD4^+^ T cells, sera, plasma, microglial cells, and whole brain	[66,267]	Sera/Plasma and GC-ICs stimulated CBE-induced GCase-targeted Mɸs	[66,67,267]
IL8 ^P++^			Sera	[127]
IL12 ^P+++^	Anti-CD3 and CD28-stimulated liver, spleen, and lung- derived CD4^+^ T cells, GC-stimulated liver, spleen, and lung derived DCs, CD4^+^ T cells, sera, and plasma	[66,267]	Sera/Plasma and GC-ICs stimulated CBE-induced GCase-targeted Mɸs	[66,67,267]
IL17 ^P+++^	Anti-CD3 and CD28-stimulated liver, spleen, and lung- derived CD4^+^ T cells, GC-stimulated liver, spleen, and lung-derived DCs, CD4^+^ T cells, β-GC 22:0 (βGL1-22)-stimulated type II natural killer+ CD19^+^ B cells, sera, and plasma	[66,67,267,268]	Sera/Plasma, peripheral blood-derived MOs, β-GC 22:0 (βGL1-22)- stimulated type II natural killer cells, CD19^+^ B cells, and GC-ICs stimulated CBE-induced GCase- targeted Mɸs	[66,67,267,268]
IL18 ^P+++^	Sera		Sera/Plasma	[127]
IL21 ^P+++^	β-GC 22:0 (βGL1-22)-stimulated type II natural killer+ CD19^+^ B cell	[67,267,268]	Peripheral blood-derived Mos, β-GC 22:0 (βGL1-22)- stimulated type II natural killer cells, and CD19^+^ B cell	[67,267,268]
IL23 ^P+++^	Anti-CD3 and CD28 stimulated liver, spleen, and lung-derived CD4^+^ T cells, GC-stimulated liver, spleen, and lung-derived DCs, CD4^+^ T cells Sera/Plasma	[66,267]	Sera/Plasma and GC-ICs stimulated CBE-induced GCase-targeted Mɸs	[66,67,267]
CCL1 ^P+++^	Sera/Plasma	[258]		
CCL2 ^M/P+++^	Sera/Plasma Lung tissues	[161,258]	Sera/Plasma, Lymph node and Spleen tissues	[67,126,161]
CCL3 ^M/P+++^	Sara/Plasma Lung tissues	[161,258]	Sera/Plasma	[67,161]
CCL4 ^P+++^	Sera/Plasma	[161,258]	Sera/Plasma	[67,161]
CCL5 ^P+++^	Sera/Plasma	[161,258]	Sera/Plasma	[67,161]
CCL6 ^M +++^	Lung tissues	[161]		
CCL9 ^M +++^	Lung tissues	[161]		
CCL17 ^M +++^	Lung tissues	[161]		
CCL18 ^P+++^	Sera/Plasma	[161]	Sera/Plasma and Spleen tissues	[67,161]
CCL22 ^M+++^	Lung tissues	[67,161]		
CXCL1 ^M/P+++^	Sera/Plasma Lung tissues	[161]		
CXCL2 ^P+++^	Sera/Plasma	[161,258]		
CXCL8 ^P+++^	Sera/Plasma	[161]	Sera/Plasma	[67,161]
CXCL9 ^M/P+++^	Sera/Plasma Lung macrophages	[121,258]		
CXCL10 ^P+++^	Sera/Plasma	[121,258]		
CXCL11 ^P+++^	Sera/Plasma	[121,258]		
CXCL12 ^M+++^	Lung tissues	[67,161]		
CXCL13 ^P+++^	Sera/Plasma	[258]		
TGFβ1 ^P+++^	Sera/Plasma	[67,161]	Lymph node	[126]
HGF ^P+++^	Sera/Plasma	[67,161]		
MCSF ^P+++^	Sera/Plasma	[67,161,258]	Sera/Plasma	[67,161]
GCSF ^P+++^	Sera/Plasma	[67,161]	Sera/Plasma	[67,161]
GMCSF ^P+++^	Sera/Plasma	[67,161,258]		

GD (Gaucher disease), C1qa (complement component 1, q subcomponent, alpha polypeptide), C3 (complement 3), C5a (complement 5a), C5a receptor1 (C5aR1), Clec4a (C-type lectin domain family 4, member d), Clec4n (C-type lectin domain family 4, member n), Clec5a (C-type lectin domain family 5, member a), Clec7a (C-type lectin domain family 7, member a), MR (Mannose receptor), IFN (interferon), TNF (tumor necrosis factor), IL (interleukin), CCL (Chemokine, C-C motif ligand), CXCL (Chemokine, C-X-C motif ligand), TGF (transforming growth factor), HGF (hepatocyte growth factor), MCSF (Mɸ colony-stimulating factor), GCSF (granulocyte colony stimulating factor), GMCSF (granulocyte-Mɸ colony stimulating factor), CBE (Conduritol B epoxide), GCase (Glucocerebrosidase), Mɸs (macrophages), iPSCs (induced pluripotent stem cells), α (alpha), β (Beta), γ (gamma), P (Protein expression level), M (mRNA expression level) + (low), ++ (moderate), and +++ (high).

**Table 5 ijms-23-14340-t005:** COVID-19 Vaccines and Alternative Treatments.

Vaccines/Alternative Treatment	Treatment Efficacy	Effectiveness Time Period, Notes	Possible Side Effects	References
mRNA Vaccines	48–>90%	5–6 months	Severe anaphylaxis, myocarditis/pericarditis, lymphadenopathy, appendicitis, herpes-zoster infection, viral facial palsy, transverse myelitis, Guillain-Barré syndrome, encephalopathy, thromboembolism, and thrombosis-thrombocytopenia syndrome	[130,132,133]
Viral vector vaccines (Sputnik)	65–91.6%			
Inactivated Vaccines	50–91.6%			
Protein-based Vaccines	90% (80% in adults over 65)			
Lopinavir and Ritonavir (protease inhibitor	No significant difference with the standard of care	NA	Could prevent the secondary immune burst without causing severe adverse events at early stages of COVID-19 and are therefore not recommended for COVID-19 at severe stages of disease	[135,136,137]
Favipiravir (Nucleoside precursor)	Improved the clinical condition of COVID-19 patients	No significant differences in hospitalization length and clinical recovery	Possible safety concerns for favipiravir include increased blood uric acid and teratogenicity, as the medication is secreted in semen and breast milk and has demonstrated toxicity in animal studies	[138,139,140,141]
Remdesivir (Nucleoside analog)	Improved recovery and clinical outcomes of hospitalized COVID-19 patients, though its effects in reducing mortality are still uncertain	Favorable influence on length of hospital stay but does not confer any mortality benefit	Nausea, hypokalemia, and headaches; mortality rate is not different from placebo	[138,139,140,142,143,144]
IFNγ	Rapid decline in SARS-CoV-2 load and a positive-to-negative viral culture conversion. Four patients recovered, and no signs of hyperinflammation were observed	Administration within 5–6 days of symptom onset/diagnosis may improve outcomes in patients with SARS-CoV-2, particularly time to viral clearance and time to clinical improvement	Nausea, temporary digestion issues	[150]
Statins	Partially associated with altered mortality	Reduced in-hospital mortality, mainly in patients with coronary disease	18% increased risk of severe COVID-19 infection	[269,270,271,272]
RAAS inhibitors	Partially associated with altered mortality	RAAS inhibitors should be continued in patients with COVID-19	Does not negatively affect clinical course of COVID-19 in hypertensive patients	[273,274]
ACE inhibitors (e.g., Lisinopril)	Efficiently treated COVID-19 vaccination- induced development of hyaluronic acid delayed inflammatory reaction	Reduced risk of COVID-19 positivity	Cough, hyperkalemia, fatigue, dizziness, headache, dysgeusia	[275,276]
IL-1α/β inhibitor (Anakinra)	Prevented severe respiratory failure of COVID-19 and decreased the mortality	Significantly reduced the risk of worse clinical outcome at day 28	Immunosuppression with anakinra may facilitate sepsis, necessitating continuous screening for bacterial superinfections	[277]
IL-6R antagonists (tocilizumab and sarilumab)	Controlled the disease severity and improved survival in critically ill patients with COVID-19	Significantly reduced the risk of worse clinical outcome at day 28	Bacterial, viral, fungal, and opportunistic infections	[278,279]
JAK inhibitors (ruxolitinib and baricitinib)	Decreased use of invasive mechanical ventilation; borderline impact on intensive care unit rates	JAK-inhibitors did not decrease length of hospitalization but exhibited a lower 28-day mortality rate	Viral (herpes, influenza), fungal, mycobacterial infections; musculoskeletal and connective tissue disorders, embolism and thrombosis, neoplasms	[280,281,282,283]
Anti-TNFα antibody (adalimumab)	Not shown; no alterations in mortality or mechanical ventilation requirements	No significant differences with the standard of care	Cutaneous swelling, pain, nose and throat sinus infections, headache, stomach pain, muscle pain	[148,284]
GMCSF (sargramostim)	Increased anti- viral antibody titers, lowered mean lung viral titers, and increased survival in SARS-CoV-2 infected human ACE2 transgenic mouse model of COVID-19	NA	Flu-like symptoms, leukocytosis and capillary leak syndrome, acute lung injury in rare cases	[285]
Anti-GMCSF monoclonal antibody (gimsilumab)	No improvement in mortality or other key clinical outcomes, i.e., pneumonia and evidence of systemic inflammation in patients with COVID-19	All-cause mortality did not vary between gimsilumab and placebo at day 43	Infections and sepsis, renal failure, infusion-related reactions	[286]
IVIGs	Improved the lung symptoms, cutaneous vasculitis, and acute encephalopathy in patients with COVID-19	IVIGs reduced mortality and increased the hospitalization length in severe COVID-19 infection	Thromboembolic events	[287,288]
Corticosteroids	Reduce mortality in patients with COVID-19	Acute trial	Hyperglycemia. No increase of neuromuscular weakness, gastrointestinal bleeding, or superinfections	[289]

SARS-CoV2 (Severe Acute Respiratory Syndrome Coronavirus-2), COVID-19 (Coronavirus Disease 2019), mRNA (messenger ribonucleic acid), IFNγ (interferon gamma), RAAS (renin-angiotensin-aldosterone system), ACE (angiotensin-converting-enzyme), IL-1α/β (interleukin-1 alpha/beta), IL-6R (interleukin-6 receptor), JAK (janus kinase), TNFα (tumor necrosis factor-α), GMCSF (granulocyte-macrophage colony stimulating factors), and IVIGs (intravenous immunoglobulins), NA (not available).

**Table 6 ijms-23-14340-t006:** Antibodies to SARS-CoV-2 in COVID-19 Patients.

SARS-CoV-2 Protein	IgG and Isotypes Antibodies to SARS-CoV-2 Proteins	IgM Antibodies to SARS-CoV-2 Proteins	IgA Antibodies to SARS-CoV-2 Proteins
S (S1 and/or S2)	IgG^++^ [260,290,291,292,293,294,295,296,297,298,299,300,301,302,303] ^+++^IgG1 > IgG2 > IgG3 [260]	IgM^+^ [260,292,293,298,299,300,301,302]	IgA^+^ [260,295,299,301,302]
RBD	IgG^++^ [251,300] and IgG1^+++^ [251]	IgM^++^ [251]	
N	IgG^+++^ [251,290,291,296,300,303,304,305,306]	IgM^+^ [293,306]	
E		IgM^+^ [292]	

SARS-CoV-2 (Severe Acute Respiratory Syndrome Coronavirus-2), COVID-19 (Coronavirus Disease 2019), S (spike), E (envelope), N (nucleocapsid protein), RBD (spike receptor binding domain), IgG (immunoglobulin G), IgM (immunoglobulin M), IgA (immunoglobulin A), + (low production), ++ (moderate production), and +++ (massive production).

### 1.6. Complement Activation Is Linked to the Increased Synthesis of Glucosylceramide Synthase Enzymes and Excess Production of Sphingolipids in COVID-19 and GD

Several studies have shown that in vivo and ex vivo stimulation of immune cells with immune complexes (e.g., IgG and IgM ICs) and complement activation products (e.g., C3, C3a, C3b, and C5a) can cause excess secretion of lysosomal enzymes (Table 7). Sphingolipids are ubiquitous components of the plasma membrane of eukaryotic cells and are essential for controlling cell proliferation, survival, and death [4,307]. However, prolonged overabundance of sphingolipids is detrimental, as seen in several lysosomal storage disorders such as GD, Fabry disease, Tay-Sachs disease, Sandhoff disease, Krabbe disease, and Niemann–Pick disease [308]. GC is an important sphingolipid, representing the backbone of more than 450 structurally different glycosphingolipids. Activation of GCS, the enzyme that places a glucosyl moiety onto ceramide, is the first pathway-committed step in the production of more complex GSLs, such as lactosylceramide and gangliosides [309].

Increased formation of GSLs such as GM2-ganglioside and lactosylceramide, have been observed upon infection by Zika virus and hepatitis C virus, respectively [310,311]. Human cytomegalovirus infection also causes an excess formation of ceramide and GM2-ganglioside [312]. Dengue virus causes the increased production of ceramide and sphingomyelin [313]. Influenza virus infection has also demonstrated increased synthesis of GC and sphingomyelin [314,315]. Conversely, suppression of the biosynthesis of cellular GSLs results in the inhibition of maturation of influenza virus particles in vitro [316,317]. It was recently suggested that GSLs support SARS-CoV-2 replication and infection [318,319]. Elevated levels of 3-ketosphinganine (16-0, 18-0, 18-1, and 20-0), sphinganine, sphingosine, dihydrosphingosine, ceramides, dihydroceramides, sphinganine-1-phosphate (S1P), GA1, and GM3 have been observed in SARS-CoV-2 infection in vitro and in vivo in experimental mouse and human cells models as well as in the sera/plasma of COVID-19 patients [1,2,3,4]. Furthermore, drugs targeting GCS inhibited the replication of SARS-CoV-2 [78].

Pathogenic variants in *GBA1/Gba1* and the resultant deficiency of acid-β glucosidase drive massive increases of GCs: 18-0, 20-0, 22-1, 22- 0, 24-1, and 24-0 in tissues (e.g., lung, liver, spleen, and brain) from mouse models and human patients with type 1, 2 and 3 GD [66,67,121,161,165,166,167,258,320,321,322,323,324,325,326]. Genetic deficiency or pharmaceutical targeting of C5aR1 in mouse models and human cellular models of GD (e.g., macrophages, and dendritic cells), showed marked reduction in the generation of GCS and GCs (e.g.,16-0, 18-0, 20-0, 22-1, 22- 0, 24-1, and 24-0) [79,258]. Importantly, studies have shown that a majority of GD patients treated with substrate reduction and/or enzyme replacement therapies and experienced SARS-CoV-2 infection only developed mild symptoms of COVID-19 and survived (Table 8). These findings suggest that the C5a–C5aR1-induced excess sphingolipid production propagates the disease in COVID-19 and GD. Thus, targeting the C5a–C5aR1 axis-mediated overproduction of GC could control SARS-CoV-2 replication and prevent development of severe disease in patients with COVID-19 and GD patients with COVID-19.

**Table 7 ijms-23-14340-t007:** Immune cell stimulation with immune complexes or compliment activation products trigger excess release of several enzymes.

Complement Activation Products	Cells	Enzyme Secretion	References
IgG and IgM-IC	PMNs ^R^	Alkaline phosphatase, Acid phosphatase, and β -Glucuronidase	[327]
BSA-anti-BSA	Mɸs ^R^	β-glucuronidase	[328]
Rheumatoid factor complex (IgG/IgM)	Leuko ^H^	β-glucuronidase	[329]
C3	Mɸs ^M^	β-glucuronidase	[330]
C3a	Mɸs ^M^	β-glucuronidase,	[330]
C3b	Mɸs ^M^	β-galactosidase, β-glucuronidase, N-acetyl-β-D-glucosaminidase	[330,331]
C3d	PMN ^H^	Elastase	[332]
C5a	PMN ^H^	β-glucuronidase, Lysozyme	[333,334]

IgG (immunoglobulin G), IgM (immunoglobulin M), IC (immune complex), C (complement), PMNs (polymorphonuclear cells), BSA (bovine serum albumin), Mɸs (macrophages), DCs (dendritic cells), Leuko (Leukocytes), M (mouse), R (rabbit), H (human).

**Table 8 ijms-23-14340-t008:** Effect of COVID-19 in Patients with Gaucher Disease.

GD Patients	Treatment	GD Patients with COVID-19	SARS-CoV-2 Positivity	Symptoms	References
181	SRT, ERT	45/181	18	Fever, dyspnea, cough, fatigue, chills, night sweats, taste and smell loss, chest pain, feeling of short breath, COVID-19 toes, extreme exhaustion, weakness, and no death	[335]
8	SRT, ERT	7/8	2	Fever, dyspnea, cough, fatigue, bilateral pneumonia, and multi-organ failure 50% death	[336]
550	NR	1	1	Mild and short clinical course managed by quarantine for 14 days	[337]
39	NR		39	Hypertension, Type 2 diabetes mellitus, dyslipidemia, asthma or chronic obstructive pulmonary disease, chronic kidney disease, coronary artery disease, heart failure, cancer	[338]
1	ERT	1	1	Fever, cough, severe neutropenia, and cavitation in the chest	[339]
1471	ERT, SRT	82	0	Asymptomatic, mildly affected but 2/82 exhibited severe/critical infection	[340]
2	ERT, Ambroxol	0	0	Mood changes, cognitive and motor deterioration	[341]

GD (Gaucher disease), COVID-19 (Coronavirus-2 Disease, 2019), SARS-CoV-2 (Severe Acute Respiratory Syndrome Coronavirus-2, SRT (Substrate Reduction Therapy), ERT (Enzyme Replacement Therapy), and NR (Not Reported).

### 1.7. Complement Activation Is Linked to the Increased Production of Chemokines, Growth Factors, Immune Cells Infiltration, and Pro-Inflammatory Cytokine Production in COVID-19 and GD

Complement activation products such as C5a are critical for generation of chemokines, growth factors, infiltration of innate and adaptive immune cells that lead to the production of pro-inflammatory cytokines and cause tissue destruction in many inflammatory diseases [200,263,264,342,343,344,345]. Findings from mouse models, cellular systems, and samples from patients with SARS-CoV-2-induced COVID-19 have shown massive generation of C5a, CCL chemokines (e.g., CCL2, CCL3, and CCL5), CXCL chemokines (e.g., CXCL9 and CXCL10), and growth factors, i.e., TGFβ, GCSF, GMCSF, VEGF, FGF, and PDGF (Table 1 and Table 2). Similar observation in mouse models, cellular systems, and human specimens from patients with GD have shown increased production of C5a and CCL chemokines (e.g., CCL1, CCL2, CCL3, CCL4, CCL5, CCL6, CCL9, CCL17, CCL18, and CCL22), CXCL chemokines (e.g., CXCL1, CXCL2, CXCL8, CXCL9, CXCL10, CXCL11, CXCL12, and CXCL13), and growth factors, i.e., TGFβ1, HGF, MCSF, GCSF, and GMCSF (Table 4).

Certain inflammatory conditions cause increased production of growth factors (e.g., GCSF, GMCSF, MCSF, HGF, VEGF, FGF, PDGF, and TGFβ1) and the indicated CCL and CXCL chemokines (i.e., CCL2/MCP1, CCL3/MIP1α, CCL4/MIP1β, CCL5/RANTES, CXCL1, CXCL2, CXCL9, CXCL10, and CXCL11), which account for the development and trafficking of immunological cells (e.g., MOs, Mɸs, DCs and the CD4^+^ and CD8^+^ T cells) from the peripheral blood and bone marrow to the sites of inflammation, where the generation of pro-inflammatory cytokines lead to tissue destruction [346,347]. A similar process is thought to occur with SARS-CoV-2 infection-induced complement activation, resulting in increased infiltration of several types of immune cells (i.e., MOs, Mɸs, DCs, PMNs, CD4^+^ T cells, CD8^+^ T cells), an overproduction of pro-inflammatory cytokines (e.g., IFN α, IFNγ, TNFα, IL1, IL2, IL6, IL7, IL8, IL12, IL17) and a decreased production of IFNα and IFNβ in COVID-19 patients (Table 1 and Table 2). In addition, the GC-induced complement activation, the increased infiltration of immunological cells (i.e., Mɸs, DCs, CD4^+^ T cells, CD8^+^ T cells) and the increased production of pro-inflammatory cytokines (i.e., IFNγ, TNFα, IL1α, IL1β, IL6, IL12, IL17,IL18, IL21, and IL23) were seen in mouse models and patients with GD (Table 3 and Table 4).

Sera and lung tissues of the MERS-CoV infected hDPP4-transgenic (hDPP4-Tg) mouse model showed increased concentrations of the C5a and C5b-9, as well as the overexpression of caspase-1 and IL1β. Furthermore, targeting C5aR1 by using anti-C5aR1 antibody treatment in the MERS-CoV-infected hDPP4-Tg mouse model caused decreased viral replication and reduced expression of IL1β and caspase-1, as well as the protection from cell death and tissue damage (e.g., lungs and spleen) [234,235]. A mouse-adapted SARS-CoV animal model showed complement activation is linked to the increased production of pro-inflammatory cytokines and chemokines, and the massive lung infiltration of inflammatory subsets of immune cells. C3^-/-^ mice infected with SARS-CoV-2 had marked reduction in the production of IL1α, IL6, TNFα, CXCL2, GCSF, and decreased lung infiltration of inflammatory subsets of immune cells (e.g., MOs and PMNs), as well as the significant protection against the development of respiratory dysfunction and lung damage when compared to control WT mice.

SARS-CoV-2 infected primary human airway epithelial cells also caused increased production of C3a, IL1α, IL6, CCL2, and CCL5 [236]. C3aR and C5aR targeting in these mice and a cell model showed pronounced reduction in the development of immune inflammation and decreased lung tissue damage [236]. Furthermore, lung, skin, and sera specimens from patients with severe COVID-19 infection have shown massive generation of C5a, C5b-C9, and C5aR1 with myeloid cells infiltration and pro-inflammatory cytokines production (Table 1). Strikingly, administration of C3 (AMY101) and C5 targeting drugs (eculizumab) resulted in marked reduction of pro-inflammatory cytokines and improved lung function in SARS-CoV-2-induced COVID-19 [80,81,82]. Similarly, GC-induced activation of complement and the resultant production of C5a caused overproduction of chemokines and growth factors, leading to increased tissue recruitment of Mɸs, DCs, and T cells. The interaction of Mɸs, DCs, and T cells eventually led to the production of inflammatory cytokines in mouse models and patients with type 1 GD [66,67,121,161]. Additionally, massive generation of chemokines (e.g., CCL2, CCL3, and CCL5), growth factors (e.g., MCSF and TGFβ), activation of the microglial cells and astrocytes, and the immense generation of pro-inflammatory cytokines (e.g., IFNγ, TNFα, IL1α, IL1β, and IL6), reactive oxygen species (ROS), nitric oxide (NO), and their combined impact inducing neuronal loss and early death were observed in *Gba*^flox/flox^; nestin-Cre mice, K14-lnl/lnl mice, 4L;C*, C57BL/6J-*Gba^tm1Nsb^* and the conduritol B epoxide-induced chemical model of nGD [167]. Type 2 and Type 3 human nGD display microglial cell activation and upregulation of pro-inflammatory cytokines (e.g., TNF α, IL1β, and IL6) that lead to the loss of neurons and to early death in patients with Type 2 and Type 3 nGD [167,172]. Furthermore, data from the *Gba1^9V/-^* mouse model, conduritol B epoxide-induced GCase-targeted mouse model and human cells, and iPSCs-derived macrophages from Type 1, Type 2 and Type 3 nGD patients, showed that the C5a–C5aR1 axis causes overgeneration of chemokines and growth factors. This leads to the increased recruitment and/or activation of innate and adaptive immune cells in visceral organs and microglial cells and/or neurons in the brain that causes overproduction of pro-inflammatory cytokines, leading to the development of severe and chronic visceral and brain tissue inflammation in GD [126]. Further, we and others have shown that genetic deficiency or pharmaceutical targeting of C5aR1 in GD mouse models and human cell model of GD resulted in marked reduction in the production of CCL chemokines (e.g., CCL1, CCL2, CCL3, CCL4, CCL5, CCL6, CCL9, CCL12, CCL17, CCL18, CCL22), CXCL chemokines (e.g., CXCL1, CXCL2, CXCL8, CXCL9, CXCL19, CXCL11, and CXCL13), and growth factors (e.g., MCSF, GCSF, and GMCSF). In addition, a decreased tissue recruitment of immune cells (e.g., Mɸs, DCs, CD4^+^ T cells), marked reduction in generation of the pro-inflammatory cytokines (e.g., IFNγ, TNFα, IL1α, IL1β, IL6, IL12p40, IL12p70, IL17,IL18, IL21, and IL23) and protection of lung, liver, and spleen disruption was de,omnstrated [79,258]. These findings suggest that targeting the C5a–C5aR1 axis could control the SARS-CoV-2/GC-induced excess production of chemokines, tissue recruitment of immune cells, pro-inflammatory cytokines production and lessen the resultant tissue damage in COVID-19 and GD patients.

## 2. Discussion

SARS-CoV-2-induced increased production of complement activation products such as C5a, C5b-C9, and C5aR, which are linked to the increased tissue recruitment and activation of MOs, Mɸs, DCs, PMNs, CD4^+^ T cells, CD8^+^ T cells, and the extensive production of pro-inflammatory cytokines, chemokines, and growth factors (Table 1 and Table 2). In addition, excess synthesis of GCS and the corresponding glycosphingolipids have been observed in COVID-19 [1,2,3,4]. Moreover, drugs targeting complement activation at the C5a–C5aR1 stage in the complement pathway and drugs targeting GCS have shown marked reduction in the replication of SARS-CoV-2, as well as the SARS-CoV-2-mediated induction of innate and adaptive immune inflammation that leads to the tissue destruction and death in patients with COVID-19 [78,80,81,82,236].

Similarly, in vitro, ex vivo, and in vivo studies performed with the *Gba1* mutant mouse models and for patients with GD have shown a link between the upregulation of C1q, C3, C4b, C5a, C5aR1 and the increased production of GC, excess recruitment and activation of innate and adaptive immune cells, and overproduction of pro-inflammatory cytokines, chemokines, and growth factors in many tissues in GD (Table 3 and Table 4). Studying the mechanistic connections between the complement-sphingolipid axis in the induction of tissue inflammation in GD, we and others have found that massive GC storage causes the development of GC-specific IgG autoantibodies that bind to GC and produce the GC-specific IgG-ICs (GC-ICs) [258,348,349]. Such ICs drive complement activation, local production of C5a, and the activation of C5aR1 in *Gba1* mutant (*Gba1^9V/-^*) mouse model and patients with GD [79,126,258]. The activation of the indicated C5a–C5aR1 axis causes the excess cellular production of GCS and GC, innate and adaptive immune cell activation, and the development of tissue inflammation in GD [79,126,258]. Furthermore, we and others have shown that genetic deficiency or pharmaceutical targeting of C5aR1 in mouse models and patients with GD reduces generation of GCS-glycosphingolipid, inhibits the production of pro-inflammatory cytokines, chemokines, and growth factors, tissue recruitment of innate and adaptive immune cells, and protects the lung, liver, and spleen tissues from damage [79,258].

Several vaccines and alternative treatments have shown poor efficacy and or immunity against SARS-CoV-2 (Table 5). Strikingly, agents targeting C5–C5a–C5aR1 or glycosphingolipid-lowering therapies have been linked to the disruption of the SARS-CoV-2 replication and the suppression of the SARS-CoV-2-induced activation of immune inflammation in COVID-19 [24,78,350,351]. However, the mechanism by which targeting C5–C5a–C5aR1 or GCS-glycosphingolipid pathways inhibit the viral replication, and the induction of immune inflammation remains unknown; thus, additional in vitro, ex vivo, and in vivo studies on different variants of SARS-CoV-2 (e.g., Alpha, Beta Delta, BA.4, and BA.5)-induced COVID-19 are required. Studies to test whether these SARS-CoV-2 variant-induced COVID-19 triggers the excess synthesis of GCS and the corresponding development of glycosphingolipids, glycosphingolipids-specific IgG antibodies, and complement activation in patients with COVID-19 and GD patients with COVID-19 are also needed. Furthermore, large scale experimental and clinical trials are critical to test whether combined targeting of complement at the level of C5a–C5aR1 and the GCS-glycosphingolipid pathway can affect the disease severity and/or death in patients with COVID-19 and in GD patients with COVID-19.

Figure 1 provides a summary of the complement–glycosphingolipid axis, as supported by study findings from patients and mouse models for COVID-19 and GD reviewed herein. It is suggested that SARS-CoV-2 infection triggers the development of SARS-CoV-2 IgM-ICs and the SARS-CoV-2 and/or GSL; GC-specific IgG1-ICs. The binding of such SARS-CoV-2 IgM-ICs to FcμR and SARS-CoV-2 IgG1-ICs/GC-specific IgG1-ICs to activating FcγR, and the direct binding of SARS-CoV-2 to mannose-binding lectin (MBL-MASP 1-2) lead to the massive production of C5a in patients with COVID-19 and GD patients with COVID-19 (Figure 1A,B). The interaction of such C5a to its C5aR1 receptor triggers GC synthase (GCS)-mediated excess synthesis of GSLs, which promotes the SARS-CoV-2 growth and replication (Figure 1C) and over production of the cytokines, chemokines, and growth factors listed in Table 1, Table 2, Table 3 and Table 4 and shown in Figure 1D. These pro-inflammatory mediators cause increased tissue infiltration and activation of several of the immune cells, i.e., MO, Mɸ, DC, PMN, and T cells, additional generation of pro-inflammatory cytokines (Figure 1E) and lead to the tissue damage in patients with COVID-19 and GD with COVID-19 (Figure 1F). Hence, targeting the C5– C5aR1–GCS glycosphingolipid pathway could protect the SARS-CoV-2-induced development of s severe form of the disease in patients with COVID-19 and GD patients with COVID-19. These findings also open up the exciting possibility of using drugs already available that effectively target the C5a–C5aR1 axis at the level of C5 (e.g., Eculizumab), which prevents C5 from being cleaved into C5a and C5b and is currently used for treating patients with Atypical hemolytic uremic syndrome and Paroxysmal Nocturnal Hemoglobinuria) [352,353]. Similarly, C5a-suppressing oral treatments (e.g., Avacopan, approved for the treatment for anti-PMN cytoplasmic antibody-associated vasculitis) and the pharmaceutical targeting of the GCS (e.g., Venglustat, which inhibits the synthesis of glycosphingolipids) can be used as potential approaches to control the tissue inflammation, organ failure, and death in patients with COVID-19 and GD patients with COVID-19 experiencing uncontrolled infections with different variants of SARS-CoV-2 (e.g., Alpha, Beta Delta, BA.4, and BA.5) [354,355].

From a clinical standpoint, this review emphasizes the possible role for medications targeting the complement-glycosphingolipid pathway as an additional anti-inflammatory therapeutic strategy to ameliorate disease progression in COVID-19 patients, patients at high-risk of developing COVID-19 or COVID-19 reinfection, and GD patients with COVID-19 disease. In sum, these immune strategies may be useful in patients for whom treatments and vaccines cannot be easily used, for example, patients at risk of developing severe, treatment-related adverse events, those with vaccine-related serious adverse events, or in patients with reinfection due to the immunity waning phenomenon.

## Figures and Tables

**Figure 1 ijms-23-14340-f001:**
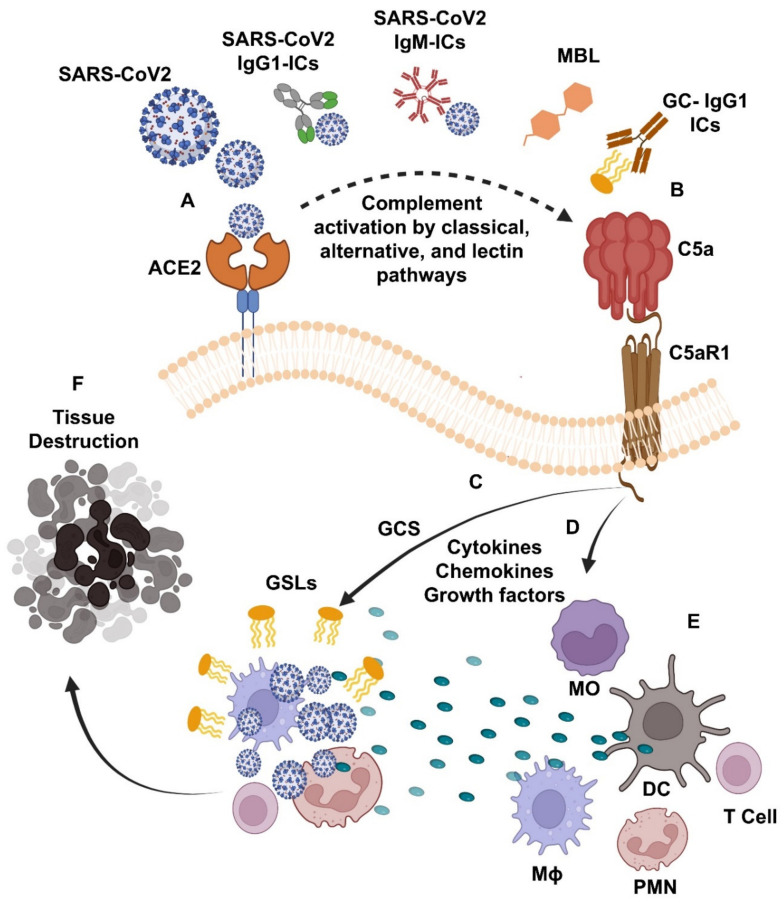
Legend. Complement-Glycosphingolipid Axis leads to the tissue inflammation in patients with Coronavirus Disease 2019 (COVID-19) and Gaucher Disease (GD). The interaction of Severe Acute Respiratory Syndrome Coronavirus-2 (SARS-CoV-2) with its cellular receptor (e.g., angiotensin converting enzyme-2; ACE2) triggers the massive generation of complement 5a (C5a) by SARS-CoV-2 and/or glycosphingolipid (GSL; glucosylceramide; GC)-specific immunoglobulin G1 immune complexes (SARS-CoV-2/GSL-IgG1-ICs)-mediated classical, SARS-CoV-2-IgM-ICs-mediated alternative, and crosslinking of SARS-CoV-2 mannose-binding lectin (MBL)-mediated lectin pathway of complement activation (**A**,**B**). The interaction of C5a and its C5aR1 receptor triggers glucosylceramide synthase (GCS)-mediated excess synthesis of GSLs, which ameliorates SARS-CoV-2 replication (**C**), as well as the overproduction of cytokines, chemokines and growth factors (**D**). These pro-inflammatory mediators cause excess tissue recruitment and activation of innate and adaptive immune cells, i.e., polymorphonuclear cell (PMN), monocytes (MO), macrophage (Mɸ), dendritic cell (DC), T cells, and the resulting excess production of pro-inflammatory cytokines (**E**; and listed in Table 1, Table 2, Table 3 and Table 4) and tissue destruction in patients with COVID-19 and GD patients with COVID-19 (**F**). Accordingly, the combined targeting of C5a–C5aR1 and/or GSL synthesis pathway could stop the SARS-CoV-2-mediated tissue inflammation and organ damage in patients with COVID-19 and GD patients with COVID-19.

## Data Availability

Not applicable.

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
