# Peer review of "Targeting the Complement–Sphingolipid System in COVID-19 and Gaucher Diseases: Evidence for a New Treatment Strategy"

_ijms, 2022, doi:10.3390/ijms232214340_

Round 1
Reviewer 1 Report
Trivedi and colleagues present an interesting review article focused on the potential for therapeutically targeting the complement-sphingolipid system based on observed evidence for the role of complement activation in tissue destruction for both COVID-19 and Gaucher Disease. The review was interesting, but I do have comments that need to be addressed prior to publication:
General critique: Is all the background on COVID-19 necessary given the scope and focus of the article? The intent is to focus on the role of the complement activation and its potential as a therapeutic target, so I’m not certain the reader benefits from several paragraphs of discussion on COVID-19, COVID-19 vaccines, alternative treatments (more relevant given the context of the review), and re-infection. It might be better to shorten the introductory sections so that you reach your main points faster.
Lines 40-41: Both instances of “alfa” should be spelled “alpha”
Lines 40-47: This is very difficult to read due to the listing of all inflammatory cytokines and growth factors the authors have interest in discussing. It would be much better to use general language: “…the increased generation of pro-inflammatory cytokines, chemokines, and growth factors,” rather than listing all of them here. They can be listed out and defined in a footnote of Table 1.
Lines 49-52: Similar critique as above – is it necessary to the reader to outline all of the symptoms associated with moderate COVID-19, or can this be limited to the primary 3-5 symptoms associated with most cases?
Line 126-128: Is the variability in vaccine efficacy related to the dominant circulating variant at the time in that country?
Lines 181-184: While true that reinfection with respiratory mucosal pathogens is much more common, the examples given for measles and mumps aren’t exactly uncommon. Measles breakthrough infections do occur (at a higher frequency than one would hope) and most mumps cases documented in the last decade have been among fully vaccinated individuals. I would recommend not naming specific pathogens here.
Line 489-502: Font size should be made uniform – several instances of larger fonts used mid-paragraph.
Author Response
Dear Reviewer,
Thank you for providing the comments on our manuscript (ID: ijms-1960430; Severe Acute Respiratory Syndrome Coronavirus 2- induced Immune Inflammation in COVID-19 is following the Law of Complement - Sphingolipid System in Gaucher Disease: Evidence of New Treatment). We were pleased to see the overall positive comments. These constructive suggestions helped us improve the quality of the manuscript. Based on the provided feedback, this manuscript has been revised extensively. Please find here our response to your comments. Hope the revised paper will be fit for the publication.
Comments:
Trivedi and colleagues present an interesting review article focused on the potential for therapeutically targeting the complement-sphingolipid system based on observed evidence for the role of complement activation in tissue destruction for both COVID-19 and Gaucher Disease. The review was interesting, but I do have comments that need to be addressed prior to publication:
General critique: Is all the background on COVID-19 necessary given the scope and focus of the article? The intent is to focus on the role of the complement activation and its potential as a therapeutic target, so I’m not certain the reader benefits from several paragraphs of discussion on COVID-19, COVID-19 vaccines, alternative treatments (more relevant given the context of the review), and re-infection. It might be better to shorten the introductory sections so that you reach your main points faster.
Response: Based on the Reviewer’s comments, the entire manuscript has been revised extensively. Furthermore, COVID-19 vaccines and alternative treatments sections (1.2 and 1.3) have been shortened and summarized in the Section 1.2 and the New Table-5
Lines 40-41: Both instances of “alfa” should be spelled “alpha”
Response: Suggested changed have been completed ion Lines 40-41 as well as in the Tables foot notes
Lines 40-47: This is very difficult to read due to the listing of all inflammatory cytokines and growth factors the authors have interest in discussing. It would be much better to use general language: “…the increased generation of pro-inflammatory cytokines, chemokines, and growth factors,” rather than listing all of them here. They can be listed out and defined in a footnote of Table 1.
Response: IFNα, IFNγ, TNFα, IL1, IL2, IL6, IL7, IL8, IL12, IL17, CCL2, CCL3, and CCL5, CXCL9 and CXCL10), TGFβ, GCSF, GMCSF, VEGF, FGF, and PDGF with the low production of IFNα and IFNβ shown in Lines 40-47 have been written in the suggested general language( pro-inflammatory cytokines, chemokines, and growth factors). They were also listed out under the foot note of the respected Tables.
Lines 49-52: Similar critique as above – is it necessary to the reader to outline all of the symptoms associated with moderate COVID-19, or can this be limited to the primary 3-5 symptoms associated with most cases?
Response: Only major symptoms of COVID-19 have been included in the lines 49-52.
Line 126-128: Is the variability in vaccine efficacy related to the dominant circulating variant at the time in that country?
Response: SARS-CoV-2 is continuously developing various variants and will continue to do so is because relatively few people globally have been vaccinated (JAMA. 2021;325(13):1241-1243. doi:10.1001/jama.2021.3370) These findings suggest that the variability in vaccine efficacy linked to the dominant circulating variant at the time in that country. These information have been included in the revision of lines 126-128.
Lines 181-184: While true that reinfection with respiratory mucosal pathogens is much more common, the examples given for measles and mumps aren’t exactly uncommon. Measles breakthrough infections do occur (at a higher frequency than one would hope) and most mumps cases documented in the last decade have been among fully vaccinated individuals. I would recommend not naming specific pathogens here.
Response: The examples given for measles, mumps, and rubella have been removed from the Lines 181-184
Line 489-502: Font size should be made uniform – several instances of larger fonts used mid-paragraph.
Response: Font sizes have been corrected throughout the manuscript.
Reviewer 2 Report
The review article "Complement - Sphingolipid System in Severe Acute 2 Respiratory Syndrome Coronavirus 2 and the Gaucher Diseases: Evidence of New Treatment" provides a comprehensive overview of common pathological features of COVID-19 and GD, and potential targeting the complement-glycosphingolipid pathway in COVID-19 patients. Although this article is appropriate for the special edition, "Complement System Entry Suspense: A Hero or Villain in Rare and Genetic Diseases," it would benefit from several corrections.
- Overall the manuscript is well written. But it needs to be thoroughly revised. For example, the abstract portion has multiple grammatical errors that the authors should fix.
- Some sections are not relevant to the topic. For example, sections 1.2 and 1.3 can be summarized on a table or mentioned briefly since these are not the focus of this article. I understand that the purpose is to highlight the treatment alternatives presented by the authors, but these sections can still be summarized. Discussion should center around areas that cover both COVID-19 and GD (1.6, 1.7).
- The introduction section should be utilized to present this article's problem and purpose. Information should be moved to other sections of the paper that specifically cover the topic. For example, information regarding COVID-19 should be included under section 1.1 so it is easier to
- The illustration should be simplified, and a figure legend should be provided.
- Correct font style and size. Variable throught the text.
Author Response
Dear Reviewer,
Thank you for providing the comments on our manuscript (ID: ijms-1960430; Severe Acute Respiratory Syndrome Coronavirus 2- induced Immune Inflammation in COVID-19 is following the Law of Complement - Sphingolipid System in Gaucher Disease: Evidence of New Treatment). Your constructive suggestions helped us improve the quality of the manuscript. Based on the provided feedback, this manuscript has been revised extensively. Hope you will like the revised paper fit for the publication.
Response to Comments:
The review article "Complement - Sphingolipid System in Severe Acute 2 Respiratory Syndrome Coronavirus 2 and the Gaucher Diseases: Evidence of New Treatment" provides a comprehensive overview of common pathological features of COVID-19 and GD, and potential targeting the complement-glycosphingolipid pathway in COVID-19 patients. Although this article is appropriate for the special edition, "Complement System Entry Suspense: A Hero or Villain in Rare and Genetic Diseases," it would benefit from several corrections.
- Overall the manuscript is well written. But it needs to be thoroughly revised. For example, the abstract portion has multiple grammatical errors that the authors should fix.
Response: Thanks for appreciating the contribution and providing the important feedback. As per the suggestions, the entire text of the revised MS has been carefully edited for spelling and grammatical errors.
- Some sections are not relevant to the topic. For example, sections 1.2 and 1.3 can be summarized on a table or mentioned briefly since these are not the focus of this article. I understand that the purpose is to highlight the treatment alternatives presented by the authors, but these sections can still be summarized. Discussion should center on areas that cover both COVID-19 and GD (1.6, 1.7).
Response: Sections 1.2 and 1.3 have been merged in the Section 1.2. Additionally, the section 1.2 and 1.3 have been summarized under the New Table-5 table and the revised discussion is more focused on the complement – sphingolipid and their effector and targeting impact in propagation of tissue inflammation in COVID-19 and GD.
- The introduction section should be utilized to present this article's problem and purpose. Information should be moved to other sections of the paper that specifically cover the topic. For example, information regarding COVID-19 should be included under section 1.1 so it is easier to
Response The Introduction has been simplified and more focused on the role of the complement activation and it’s potential as a therapeutic target in COVID-19. Several of the information related to COVID-19 has been moved from Introduction to section 1.1
- The illustration should be simplified, and a figure legend should be provided.
Response: The illustration has been modified and figure legend has been included
- Correct font style and size. Variable through the text.
Response: Font sizes have been corrected throughout the manuscript.
Round 2
Reviewer 2 Report
The manuscript is relevant and summarizes valuable information for researchers and practitioners interested in this area.
The authors incorporated most suggestions, and this manuscript will be ready for publication after a few additional edits.
- Authors should do a final review to ensure proper grammar.
- I did not appreciate any changes to the figure indicated by the authors. Can the authors highlight these changes?
- Changes to the title are confusing. Is there a rationale behind this? Can it be further clarified since we are discussing the implications of the sphingolipid system in both diseases?
Author Response
Dear Reviewer,
Thank you for providing the positive comments on our manuscript (ID: ijms-1960430). Based on the reviewer’s additional suggestions, this manuscript has been revised broadly. Please find below the pointwise responses.
Reviewer‘s Comments:
The manuscript is relevant and summarizes valuable information for researchers and practitioners interested in this area.
The authors incorporated most suggestions, and this manuscript will be ready for publication after a few additional edits.
- Authors should do a final review to ensure proper grammar.
Response: MS has been reasonably revised for grammar.
- I did not appreciate any changes to the figure indicated by the authors. Can the authors highlight these changes?
Response: Sorry for missing addressing the previous suggestions for the Figure. Based on the Reviewer’s comments, figure has been simplified (See the Revised Figure 1).
- Changes to the title are confusing. Is there a rationale behind this? Can it be further clarified since we are discussing the implications of the sphingolipid system in both diseases?
Response: Title has been changed.